# High Incidence of Metastatic Infections in Panton-Valentine Leucocidin-Negative, Community-Acquired Methicillin-Resistant *Staphylococcus aureus* Bacteremia: An 11-Year Retrospective Study in Japan

**DOI:** 10.3390/antibiotics12101516

**Published:** 2023-10-06

**Authors:** Hitoshi Kawasuji, Yoshihiro Ikezawa, Mika Morita, Kazushige Sugie, Mayu Somekawa, Masayoshi Ezaki, Yuki Koshiyama, Yusuke Takegoshi, Yushi Murai, Makito Kaneda, Kou Kimoto, Kentaro Nagaoka, Hideki Niimi, Yoshitomo Morinaga, Yoshihiro Yamamoto

**Affiliations:** 1Department of Clinical Infectious Diseases, Toyama University Graduate School of Medicine and Pharmaceutical Sciences, Toyama 930-0194, Japan; 2Department of Clinical Laboratory and Molecular Pathology, Toyama University Graduate School of Medicine and Pharmaceutical Sciences, Toyama 930-0194, Japan; 3Department of Microbiology, Toyama University Graduate School of Medicine and Pharmaceutical Sciences, Toyama 930-0194, Japan

**Keywords:** methicillin-resistant *Staphylococcus aureus* (MRSA), healthcare-associated MRSA (HA-MRSA), community-associated MRSA (CA-MRSA), staphylococcal cassette chromosome *mec* (SCC*mec*), bacteremia, *cna*, *fnbB*

## Abstract

Panton-Valentine leucocidin (PVL)-negative community-associated methicillin-resistant *Staphylococcus aureus* (CA-MRSA) was originally disseminated in Japan and has since replaced healthcare-associated MRSA (HA-MRSA). However, the clinical characteristics of CA-MRSA bacteremia (CA-MRSAB) compared with those of HA-MRSA bacteremia (HA-MRSAB) are unknown. We aim to clarify differences and investigate associations between the clinical manifestations and virulence genes associated with plasma-biofilm formation in PVL-negative CA-MRSA. From 2011 to 2021, when CA-MRSA dramatically replaced HA-MRSA, 79 MRSA strains were collected from blood cultures and analyzed via SCC*mec* typing and targeted virulence gene (*lukSF-PV*, *cna*, and *fnbB*) detection. The incidence of metastatic infection was significantly higher in CA-MRSAB than in HA-MRSAB. PVL genes were all negative, although *cna* and *fnbB* were positive in 55.6% (20/36) and 50% (18/36) of CA-MRSA strains and 3.7% (1/27) and 7.4% (2/27) of HA-MRSA strains, respectively. *cna* and *fnbB* carriage were not associated with the development of metastatic infections in MRSAB; however, the bacteremia duration was significantly longer in CA-MRSAB harboring *cna*. CA-MRSAB may be more likely to cause metastatic infections than HA-MRSAB. Since CA-MRSA is dominant in Japan, suspected metastatic infection foci should be identified by computed tomography, magnetic resonance imaging, and echocardiography when treating MRSAB.

## 1. Introduction

Methicillin-resistant *Staphylococcus aureus* (MRSA), one of the most common causes of bacteremia, leads to high mortality in community and hospital settings.

From its first reports in 1961 to the 1990s, MRSA infections were commonly associated with healthcare contact [1]. In the 1990s, MRSA infection cases began occurring in individuals without prior hospitalization, leading to separate definitions of healthcare-associated MRSA (HA-MRSA) and community-associated MRSA (CA-MRSA) [1]. However, from the 1990s onwards, CA-MRSA clones were rapidly disseminated in healthcare settings, and it is no longer difficult to distinguish between them by site of acquisition [2,3,4].

CA-MRSA typically harbors staphylococcal cassette chromosome *mec* (SCC*mec*) types IV and V and has different characteristics from HA-MRSA harboring SCC*mec* I, II, and III. CA-MRSA strains are more susceptible to antimicrobial agents such as fluoroquinolones, macrolides, lincosamides, and aminoglycosides [5]. In Japan, most CA-MRSA strains differ from those in Europe and the United States [1,6]. The USA300 clone, a representative clone of CA-MRSA that carries the Panton-Valentine leukocidin (PVL) gene (*lukSF-PV*), has spread throughout the United States and is now endemic globally [1,6]. Outside the United States, the Southwest Pacific clone (Sequence type [ST]30/SCC*mec* IV) around the world, the Taiwan clone (ST59/SCC*mec* V) in South Asia, and the European clone (ST80/SCC*mec* IV) in Europe are the dominant clones and all these CA-MRSAs carry PVL genes [7]. On the other hand, a recent national surveillance study of MRSA bacteremia (MRSAB) in Japan showed that CA-MRSA seldom carries PVL genes; the carriage of the *lukSF-PV* in CA-MRSA is < 6.0% [5].

A previous Japanese study reported that CA-MRSA was increasing in bloodstream infections between 2015 and 2017 [8] and, by 2019, had replaced HA-MRSA as the dominant strain [5]. The clinical characteristics of Japanese CA-MRSA infections may differ from those in the United States and Europe; however, national surveillance studies of MRSAB in Japan have not assessed the association between molecular characteristics and clinical manifestations due to difficulties in collecting clinical information [5].

A recent study reported the higher plasma biofilm formation ability of SCC*mec* type IV clones than that of SCC*mec* type II clones. This ability was particularly pronounced in strains harboring *cna* or *fnbB* [8]. Cna and FnbB are involved in adherence to many cells or tissues and have been reported to be implicated in invasive infections, such as sepsis and septic arthritis, in animal models [9,10,11].

However, among patients with MRSAB in real-world clinical practice, whether genetic characteristics such as CA-MRSA or strains harboring *cna* and *fnbB* contribute to disease pathogenesis in MRSAB remains unknown.

We aim to clarify the differences in clinical characteristics between HA-MRSA bacteremia (HA-MRSAB) and CA-MRSA bacteremia (CA-MRSAB) in a Japanese tertiary care hospital and investigate associations between the clinical manifestations and virulence genes of PVL-negative CA-MRSA, which is dominant in Japan.

## 2. Results

### 2.1. Study Population

At Toyama University Hospital, MRSA was identified in 22% to 31% of clinical *S. aureus* isolates during the study period (Appendix A). A total of 79 strains isolated from 79 patients with clinically significant MRSAB were included in this analysis. Of these strains, 25.3% (*n* = 20) were classified as SCC*mec* type II, 8.9% (*n* = 7) as SCC*mec* type III, 40.5% (*n* = 32) as SCC*mec* type IV, 5.1% (*n* = 4) as SCC*mec* type V, 10.1% (*n* = 8) as SCC*mec* type VI, 1.3% (*n* = 1) as SCC*mec* type VIII, and 8.9% (*n* = 7) as non-typable (Figure 1A). SCC*mec* typing by year showed that the prevalence of SCC*mec* types II and III, classified as the HA-MRSA genotype, decreased dramatically from 80 to 100% in 2011–2012 to 0–10% in 2018–2021 (Figure 1B). Instead, CA-MRSA strains carrying SCC*mec* types VI and V replaced HA-MRSA and became dominant in the later period (80–100% in 2018–2019). In 2020–2021, some strains were non-typable. All strains were *mecA*-positive but also *lukSF-PV*-negative.

### 2.2. Clinical Characteristics and Outcomes of HA-MRSAB and PVL-Negative CA-MRSAB

The demographic and clinical characteristics of patients with CA-MRSAB and HA-MRSAB are shown in Table 1. The percentage of patients with the hospital onset (HO) of bacteremia was significantly higher in the HA-MRSAB group than in the CA-MRSAB group; however, 22/36 (61.1%) patients with CA-MRSAB had hospital-onset. Age, gender, BMI, comorbidity, the Charlson comorbidity index, presence of foreign bodies, and disease severity were not significantly different between the HA-MRSAB and CA-MRSAB groups. Overall, the most frequent source of infection was catheter-related bloodstream infection (CRBSI) (44.3%), followed by lower respiratory tract infection (21.5%). MRSAB caused by surgical site infections was significantly more frequent in HA-MRSAB (25.9%) than in CA-MRSAB (0.0%), whereas MRSAB caused by skin and soft tissue infections was significantly more frequent in CA-MRSAB (36.1%) than in HA-MRSAB (3.7%).

Regarding antimicrobial therapy, the frequency of inappropriate empirical therapy, initial anti-MRSA drugs, the removal of intravascular devices within 5 days after bacteremia onset in patients with CRBSI, and the performance of follow-up blood cultures did not differ significantly between the two groups. The all-cause 30-day and in-hospital mortality rates also did not differ significantly between HA-MRSAB and CA-MRSAB.

All clinical characteristics, except for the source of infection, did not differ significantly between HA-MRSA and CA-MRSA, and the incidence of metastatic infection was significantly higher in CA-MRSA than in HA-MRSA. Details of the metastatic infections are presented in Appendix A.

### 2.3. Prevalence of cna and fnbB Genes and Their Relationships with Metastatic Infections in PVL-Negative CA-MRSAB

The prevalence of *cna* and *fnbB* among the SCC*mec* types is shown in Table 2. *cna* and *fnbB* positivity was observed in 55.6% (20/36) and 50% (18/36) of the CA-MRSA strains, and 3.7% (1/27) and 7.4% (2/27) of the HA-MRSA strains, respectively. Among CA-MRSAB, the carriage of *cna* and *fnbB* was not associated with the development of metastatic infections (Appendix A).

### 2.4. Associations between Clinical Characteristics and Virulence Genes in PVL-Negative CA-MRSAB

We then sought to determine the differences in clinical characteristics between CA-MRSAB isolates carrying *cna* or *fnbB* and those that did not. Among patients from whom follow-up blood cultures were obtained within 14 days after onset, although the source of infection, presence of metastatic infections, comorbid medical conditions, including the presence of a foreign body, severity, treatment, or interval between repeated blood culture collections did not differ (Appendix A), the bacteremia duration was significantly longer in patients with bacteremia caused by CA-MRSA harboring *cna* than in those without (median day [range]: 1 [1–15] vs. 1 [1–4], *p* = 0.043) (Figure 2A). By contrast, the bacteremia duration did not differ significantly between patients with bacteremia caused by CA-MRSA harboring *fnbB* and those who did not (median days [range]: 1 [1–15] vs. 1 [1–11], *p* = 0.12). The prevalence of CA-MRSA strains harboring *cna* increased from 27.3% (3/11) in 2011–2017 to 68.0% (17/25) in 2018–2021 (Figure 2B).

## 3. Discussion

This study’s results show that the dominant bacteremia-causing clone in our hospital changed from HA-MRSA to CA-MRSA during the study period, and this finding is consistent with those of previous studies [5,8,12]. However, because an increasing number of CA-MRSA isolates are unique and differ from those in other countries [5], there are limited reports on the relationship between the genetic characteristics and clinical presentation of MRSAB.

The results of the present study show that CA-MRSAB causes metastatic infections, including infective endocarditis, vertebral osteomyelitis, iliopsoas abscess, septic arthritis, ocular infection, and septic pulmonary embolism, more frequently than HA-MRSAB. Although the presence of a permanent foreign body, hemodialysis, higher Charlson comorbidity score, and inappropriate antimicrobial treatment are reported as predictive factors for the development of metastatic infections in *S. aureus* bacteremia [13], the results of the present study showed no significant difference in these factors between CA-MRSAB and HA-MRSAB. Although this univariate analysis lacked statistical power owing to the insufficient number of patients, the results suggest the need for careful attention to detect metastatic infections via computed tomography or the magnetic resonance imaging of suspected foci, along with echocardiography, in CA-MRSAB.

The characteristics of MRSA infections include biofilm formation, which leads to the spread of bacteria into the bloodstream, causing sepsis and metastasis [14]. A previous study measuring biofilm production in 83 MRSA isolates from human infections reported that isolates from bacteremia showed a higher percentage of biofilm formation (80.5%) than those from diabetic foot infections (77.6%) and osteomyelitis (58.3%) [15]. In addition, a previous report showed that *S. aureus* forms mature biofilms in the presence of blood plasma [14]. A recent study by the same group demonstrated the higher plasma biofilm-forming ability of SCC*mec* type IV clones than thise of the SCC*mec* type II clone, and the strains that possessed *cna* and *fnbB* exhibited an even higher plasma biofilm-forming ability [8].

A previous study that conducted whole-genome sequencing (WGS) and the cluster analysis of 154 MRSA strains showed that strains in a specific cluster that were more likely to cause bacteremia were all positive for *cna* [16]. Cna is a cell-wall-anchored protein that participates in adhesion to collagen-rich tissues. Cna also prevents complement fixation and likely contributes to innate immune evasion during infection [17]. Cna has been studied as an *S. aureus* virulence-determining factor, and previous studies comparing wild-type and *cna*-positive mutant strains have demonstrated that *cna*-positive strains cause considerably more septic arthritis symptoms in a mouse model and outnumber the mutant strain 24 h after inoculation in a rat model of infective endocarditis [18]. Associations between *cna* and bacteremia, infective endocarditis, and osteomyelitis have also been reported in several clinical studies [16,19]. Furthermore, MRSA isolates causing persistent bacteremia carry *cna* at a higher rate than that in resolving bacteremia [20]. Consistently, in the present study, the duration of MRSAB was significantly longer in patients with bacteremia caused by CA-MRSA harboring *cna* than in those without.

By contrast, we observed no correlations between *fnbB* and persistent bacteremia or between the presence of *cna* or *fnbB* and the development of metastatic infections in CA-MRSAB. A previous study demonstrated that the detachment of biofilm cells in the early phase of biofilm formation resulted in the colonization of distant host sites and metastatic infections [21,22]. Based on the results of this previous study and ours, the detachment from biofilms seems unlikely to occur in CA-MRSA strains harboring the *cna* or *fnbB* adhesion genes. Indeed, a recent study reported a higher ratio of detached and attached bacterial numbers in CA-MRSA with mutant *cna* than the normal strain, suggesting that these factors are a strong predictor of the incidence of bacteremia [16].

No previous studies have reported differences in clinical manifestations between PVL-negative CA-MRSAB and HA-MRSAB in Japan. We found that CA-MRSAB more frequently caused metastatic infections compared to HA-MRSAB. Furthermore, among cases with CA-MRSAB, the presence of *cna* and *fnbB* was not associated with the development of metastatic infections; however, *cna* was associated with a longer MRSAB duration.

Recent data have revealed that CA-MRSA tends to show a low minimum inhibitory concentration (MIC) of carbapenems, cephems, minocycline, and clindamycin [5]. However, these differences are insufficient to distinguish between CA-MRSA and HA-MRSA. To identify CA-MRSA and HA-MRSA more accurately, the genotyping of SCC*mec* is required. In the present study, we focus on clinical significance and applicability to clinical practice; therefore, we performed PCR-based assays commonly used in hospital laboratories.

Our study has several limitations. First, multi-locus sequence typing and whole-genome sequencing of the strains were not performed; thus, factors other than SCC*mec* types and *cna* may affect clinical manifestations such as metastatic infections or persistent bacteremia. Second, we did not examine the amount of toxins produced. Finally, this was a single-center retrospective study with an insufficient number of enrolled patients.

## 4. Materials and Methods

### 4.1. Collection of Clinical Isolates

Between 2011 and 2021, MRSA strains were isolated from the blood cultures of 79 different adult patients (age ≥ 18 years) at Toyama University Hospital, a 612-bed tertiary-care teaching hospital in Japan. Genetic analyses, including SCC*mec* typing and virulence gene detection, were performed on the 79 stored MRSA strains.

MRSAB cases were identified using microbiological laboratory records. We included only clinically significant cases, defined as the occurrence of at least one positive blood culture for MRSA in the presence of concomitant signs of infection, as described previously [19]. In patients with more than one episode of MRSAB during the study period, only the first episode was included [23], and the strain isolated from the first positive blood culture was genetically analyzed. Episodes of polymicrobial bacteremia in which MRSA was isolated from a single blood culture bottle and when the patient received inappropriate empirical treatment for the co-pathogen and episodes of breakthrough MRSAB were excluded [23]. Additionally, patients who died within 48 h after the extraction of the first set of blood cultures and those receiving palliative care were excluded, as described previously [23].

### 4.2. Molecular Characterization

Each MRSA strain was cultured aerobically on trypticase soy agar (Becton Dickinson [BD], Sunnyvale, CA, USA) at 37 °C overnight. Bacterial DNA was extracted from colonies using a modified version of a previously described simple alkali wash and heat method [24]. Briefly, a loop of bacteria scraped from a single colony was suspended in 50 μL of NaOH (0.05 mol/L), and the mixture was incubated at 95 °C for 10 min before 11 μL of Tris-HCI buffer (pH 7.0) was added. One microliter of a crude DNA extract diluted 10-fold with distilled water was used as the template for genetic analysis.

MRSA SCC*mec* types were determined using a polymerase chain reaction (PCR)-based assay, as described previously [25]. Five PCR primer sets, *mecA*, *ccrB2*, *mecI*, *IS1272J*, *ccrC*, and *mecC2*, were used to distinguish SCC*mec* types I–IV and VIII [25]. Strains not identified as SCC*mec* types I–VI and VIII were classified as non-typable.

Staphylococcal virulence genes were detected via a PCR using previously reported primers [26,27]. The target virulence genes included the PVL-encoding (*lukSF-PV*), *cna*, and *fnbB* genes. Four strains of MRSA, ATCC BAA-1708 (SCC*mec* type II), NCTC 13142 (SCC*mec* type IV), BAA-2094 (SCC*mec* type V), and NTCC 14245 (USA 300 clone) were used as positive controls for SCC*mec* typing and virulence gene detection analysis.

### 4.3. Clinical Characterization

We retrospectively reviewed the electronic medical records of all patients with MRSAB for demographic and clinical data, including age, gender, body mass index (BMI), the acquisition of infection, comorbid medical conditions, Charlson comorbidity index, the presence of foreign bodies, markers of severity (septic shock, mechanical ventilation, Pitt’s bacteremia score, Sequential Organ Failure Assessment [SOFA] score, and quick SOFA score) on the day of index blood culture collection, the primary focus of infection, metastatic foci and complications, the number of positive blood cultures, and isolated microorganisms.

For each patient, the time of bacteremia onset was defined as the time when the first sample was obtained when the blood culture result was positive. The acquisition locations were classified as community-acquired (CA), healthcare-associated (HCA), or HO, as defined previously [28]. Septic shock was diagnosed according to current guidelines [29]. In cases where blood gas analyses were not performed, SOFA scores were calculated using the following equation [30]:[SpO_2_/FiO_2_ = 64 + 0.84 × (PaO_2_/FiO_2_)]

The outcome variables included bacteremia duration, all-cause 30-day mortality, and in-hospital mortality. The duration of bacteremia was calculated as the number of days between the first and last positive blood cultures for MRSA [31]. Patients without repeat positive blood cultures and who experienced clinical success were categorized as having one day of bacteremia [31]. Persistent bacteremia was defined as lasting for two days or more despite active antibiotic therapy [32]. Repeated blood cultures were drawn at the discretion of treating physicians. Patients were followed up until death, withdrawal after discharge, or transfer to another healthcare facility.

### 4.4. Statistical Analysis

The participant’s medical and demographic characteristics were summarized as medians (interquartile ranges) or numbers (percentages). Differences between the two groups were tested using the Mann–Whitney U or Fisher’s exact tests. Statistical significance between different groups was defined as *p* < 0.05. Statistical analysis and figure construction were performed using JMP Pro version 17.0.0 software (SAS Institute Inc., Cary, NC, USA) and GraphPad Prism version 9.5.1 (GraphPad Software, San Diego, CA, USA).

### 4.5. Ethics Approval

This study was performed according to the principles of the Declaration of Helsinki and was approved by the ethical review board of the University of Toyama (approval No.: R2019140). The requirement for written informed consent was waived for this retrospective study by the ethical review board.

## 5. Conclusions

In conclusion, the results of the present study show that metastatic infections occurred more frequently in patients with PVL-negative CA-MRSAB than in those with HA-MRSAB. In addition, the carriage of *cna* was associated with a longer MRASAB duration. These findings indicate that bacterial genetic characteristics are associated with the clinical characteristics of MRSAB. Since the prevalence of CA-MRSA has been increasing and has already replaced HA-MRSA in Japan, there is a need for careful attention to detect metastatic infections by performing computed tomography or magnetic resonance imaging for suspected foci, along with echocardiography during the treatment of MRSAB.

## Figures and Tables

**Figure 1 antibiotics-12-01516-f001:**
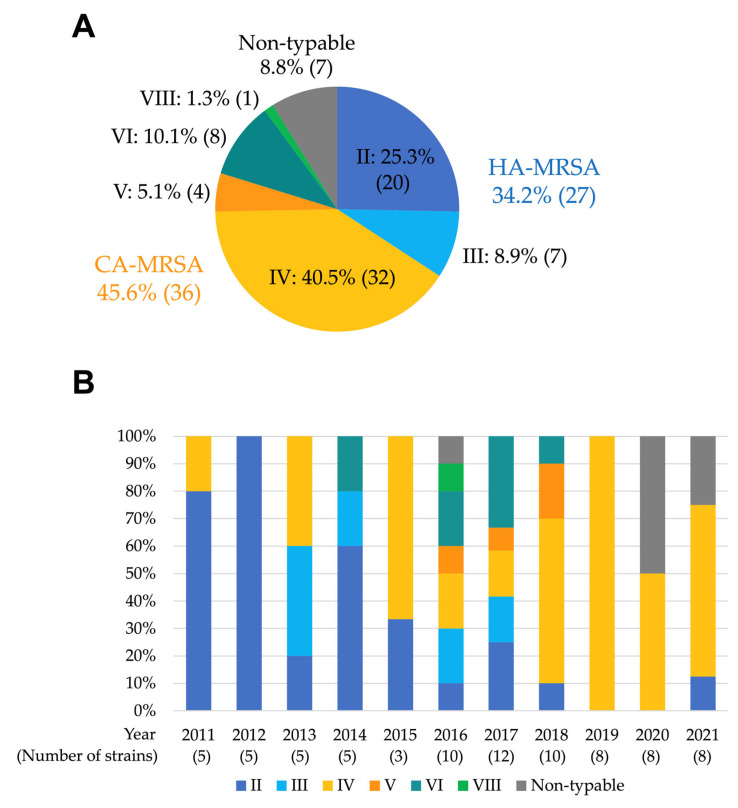
Staphylococcal cassette chromosome *mec* (SCC*mec*) typing of methicillin-resistant *S. aureus* (MRSA) isolates from blood cultures at Toyama University Hospital. (**A**) Proportions and (**B**) 11-year trends in SCC*mec* types.

**Figure 2 antibiotics-12-01516-f002:**
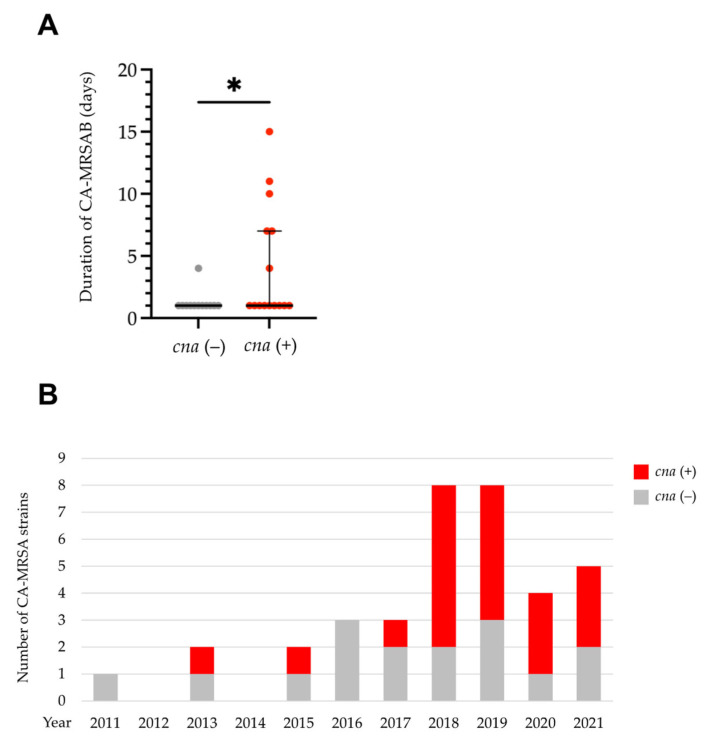
Clinical characteristics and annual trends of community-acquired methicillin-resistant *S. aureus* (CA-MRSA) isolates carrying *cna*. (**A**) The presence of *cna* was associated with a longer duration of CA-MRSAB. Each dot represents an individual patient. (**B**) Eleven-year trend in the number of CA-MRSA isolates carrying *cna*. * *p* < 0.05. Bars: medians with interquartile ranges.

**Table 1 antibiotics-12-01516-t001:** Demographic and clinical characteristics of patients with CA-MRSAB and HA-MRSAB.

Characteristics	Total, *n* = 79	HA-MRSAB, *n* = 27 (34.2%)	CA-MRSAB, *n* = 36 (45.6%)	*p* Value
Age, years, median (IQR)	72 (65–81)	77 (68–82)	71 (63.3–80.5)	0.14
Sex, male, n (%)	54 (68.4)	17 (63.0)	26 (72.2)	0.59
BMI (kg/m^2^), median (IQR)	20.1 (17.6–22.8)	20.1 (18.0–22.7)	20.6 (17.8–22.8)	0.81
Acquisition of infection, n (%)				
Hospital-onset	57 (72.1)	23 (85.2)	22 (61.1)	<0.01
Source of infection, n (%)				
CRBSI	35 (44.3)	9 (33.3)	15 (41.7)	0.60
Lower respiratory tract	17 (21.5)	5 (18.5)	10 (27.8)	0.55
Surgical site	7 (8.9)	7 (25.9)	0 (0.0)	<0.01
Infective endocarditis	4 (5.1)	0 (0.0)	3 (8.3)	0.25
Intra-abdominal	5 (6.3)	2 (7.4)	2 (5.6)	1.00
Skin and soft tissue	17 (21.5)	1 (3.7)	13 (36.1)	<0.01
Urinary tract	10 (12.7)	3 (11.1)	4 (11.1)	1.00
Mediastinitis	3 (3.8)	3 (11.1)	0 (0.0)	0.07
Bone and joint	10 (12.7)	3 (11.1)	4 (11.1)	1.00
Other and unknown	2 (2.5)	2 (7.4)	0 (0.0)	0.18
Metastatic infection, n (%)	18 (22.8)	2 (7.4)	12 (33.3)	0.017
Polymicrobial bacteremia, n (%)	8 (10.1)	1 (3.7)	5 (13.9)	0.23
Comorbid medical conditions, n (%)				
Malignancy	35 (44.3)	14 (51.9)	15 (41.7)	0.45
Diabetes mellitus	20 (25.3)	7 (25.9)	10 (27.8)	1.00
Chronic heart failure	8 (10.1)	3 (11.1)	5 (13.9)	1.00
Valvular heart disease	3 (3.8)	2 (7.4)	1 (2.8)	0.57
Chronic renal failure	14 (17.7)	5 (18.5)	7 (19.4)	1.00
Hemodialysis	5 (6.3)	0 (0.0)	5 (13.9)	0.065
Cirrhosis	1 (1.2)	0 (0.0)	0 (0.0)	–
Chronic pulmonary disease	11 (13.9)	5 (18.5)	4 (11.1)	0.48
Cerebrovascular event	4 (5.1)	2 (7.4)	2 (5.6)	1.00
Burn injury	2 (2.5)	0 (0.0)	2 (5.6)	0.50
Foreign body ^a^	18 (22.8)	8 (29.6)	6 (16.7)	0.24
Immunosuppression ^b^	13 (16.4)	4 (14.8)	5 (13.9)	1.00
Charlson comorbidity index, median (IQR)	3 (2–4)	3 (2–4)	3 (2–4)	0.64
Severity of infection				
Septic shock, n (%)	10 (12.7)	1 (3.7)	7 (19.4)	0.12
Mechanical ventilation, n (%)	5 (6.3)	1 (3.7)	4 (11.1)	0.38
Pitt’s bacteremia score at onset of bacteremia, median (IQR)	0 (0–2)	0 (0–1)	0.5 (0–2)	0.11
SOFA score at onset of bacteremia, median (IQR)	2 (0–4)	1 (0–4)	3 (0–5)	0.18
Quick SOFA score at onset of bacteremia, median (IQR)	1 (0–1)	1 (0–1)	1 (0–1)	0.85
Antimicrobial therapy, n (%)				
Inappropriate empirical therapy	62 (78.5)	20 (74.1)	31 (86.1)	0.33
Reduced susceptibility to vancomycin, n (%) MIC ≥2.0 mg/L	16 (20.3)	11 (40.7)	2 (5.6)	<0.01
Initial anti-MRSA therapy, n (%)				
Vancomycin	36 (45.6)	11 (40.7)	19 (52.8)	0.45
Teicoplanin	17 (21.5)	7 (25.9)	5 (13.9)	0.33
Arbekacin	1 (1.3)	0 (0.0)	1 (2.8)	1.00
Linezolid	13 (16.5)	3 (11.1)	7 (19.4)	0.49
Daptomycin	9 (11.4)	4 (14.8)	3 (8.3)	0.45
Management, n (%)				
Removal of intravascular device within 5 days after bacteremia onset in patients with CRBSI	34 (89.5)	9/9 (100)	15/17 (88.2)	0.53
Follow-up blood culture	65 (82.3)	18 (66.7)	32 (88.9)	0.057
Anti-MRSA therapy duration ≥14 days	61 (84.8)	21/24 (87.5)	31/34 (91.2)	0.68
Anti-MRSA therapy duration, median (IQR)	22 (14–36)	23 (14–35)	25 (15.3–46)	0.31
Outcome, n (%)				
Persistent bacteremia	16 (24.6)	3 (16.7)	8 (25.0)	0.72
All-cause 30-day mortality	9 (11.4)	4 (14.8)	3 (8.3)	0.45
All-cause in-hospital mortality	18 (22.7)	7 (25.9)	8 (22.2)	0.77
Length of hospital stay, median (IQR)	55 (28–114)	99 (67–140)	44.5 (28.3–106)	0.012

^a^ Including prosthetic valve, vascular graft, joint, or pacemaker. ^b^ Consisted of transplantation, AIDS, and use of corticosteroids or nontransplant immunosuppressive medications. HA-MRSAB, healthcare-associated methicillin-resistant *S. aureus* bacteremia; CA-MRSAB, community-associated methicillin-resistant *S. aureus* bacteremia; BMI, body mass index; CRBSI, catheter-related bloodstream infection; MIC, minimum inhibitory concentration; IQR, interquartile range; SOFA, Sequential Organ Failure Assessment; MRSA, methicillin-resistant *S. aureus*.

**Table 2 antibiotics-12-01516-t002:** Rates of *cna* and *fnbB* identification by SCC*mec* type.

SCC*mec* Type, *n* (%)	*cna* (+)	*fnbB* (+)
II	1/20 (5.0)	1/20 (5.0)
III	0/7 (0.0)	1/7 (14.3)
IV	18/32 (34.3)	18/32 (34.3)
V	2/4 (50.0)	0/4 (0.0)
VI	5/8 (62.5)	2/8 (25.0)
VIII	0/1 (0.0)	0/1 (0.0)
Non-typable	6/7 (85.7)	4/7 (57.1)

SCC*mec*, Staphylococcal cassette chromosome *mec.*

## Data Availability

All data are provided in the manuscript and Appendix A.

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
