# Peer review of "High Incidence of Metastatic Infections in Panton-Valentine Leucocidin-Negative, Community-Acquired Methicillin-Resistant Staphylococcus aureus Bacteremia: An 11-Year Retrospective Study in Japan"

_antibiotics, 2023, doi:10.3390/antibiotics12101516_

Round 1

Reviewer 1 Report

The study is interesting in that clinicians could be helped in managing bacteriemia after genetic characterization of the strains of S. aureus. However, the role of the cna gene was already known and so the contribution of the presented results seems to be minor. This is my main concern regarding the paper.

I'm surprised of the limitations of the study described by the authors in the Discussion, perhaps they could calculate the statistical significance of the data with the studied patients and strains. I wonder why they didnt solve the problem of the sequencing and production of toxins if they consider are important aspects conditioning the results. 

The explanation of the cna and fnbB encoding activity and the known role of the genes have to be give in the first part of the study, even in the Introduction.

Table 3 seems not to be needed

Reviewer 2 Report

The authors present an interesting case report highlighting the higher incidence of infection in CA-MRSABs compared with HA-MRSABs. The descriptive statistical methodology is consistent with the objectives of the study. Although it has the limitations of a retrospective study, it may be of considerable interest for future prospective studies.

Author Response

We are grateful for the valuable comments.

Reviewer 3 Report

Beyond the limitations that already have been mentioned in the last paragraph of the discussion, mainly the lack of whole genome sequencing in order to reveal factors that might have affected clinical findings, I find the study interesting and well conducted. 

Author Response

Thank you for your valuable comments. 

Reviewer 4 Report

This study analyzed 10 years of CA-MRSA in Japan including  SCCmec typing, targeted virulence genes, metastatic infection and clinical characteristics. The study was interesting and meaningful. It was suggested to be accepted after minor revision.

Main Concerns:
(1) What’s the difference of CA-MRSA in Japan with other countries.

(2) Did some of these strains in this study obtained genome sequences? I think the phylogenetic analysis based on genome is also very important.

(3) The resistance genes could also identified in the genome.

(4) Is there any way to distinguish between CA-MRSA and HA MRSA?

Round 2

Reviewer 1 Report

Ther modifications the authors produced as suggested by me  are OK

English is OK